# Learning to Ascend Stairs and Ramps: Deep Reinforcement Learning for a Physics-Based Human Musculoskeletal Model

**DOI:** 10.3390/s22218479

**Published:** 2022-11-03

**Authors:** Aurelien J. C. Adriaenssens, Vishal Raveendranathan, Raffaella Carloni

**Affiliations:** Bernoulli Institute for Mathematics, Computer Science and Artificial Intelligence, Faculty of Science and Engineering, University of Groningen, Nijenborgh 9, 9747 AG Groningen, The Netherlands

**Keywords:** deep reinforcement learning, computer simulation

## Abstract

This paper proposes to use deep reinforcement learning to teach a physics-based human musculoskeletal model to ascend stairs and ramps. The deep reinforcement learning architecture employs the proximal policy optimization algorithm combined with imitation learning and is trained with experimental data of a public dataset. The human model is developed in the open-source simulation software OpenSim, together with two objects (i.e., the stairs and ramp) and the elastic foundation contact dynamics. The model can learn to ascend stairs and ramps with muscle forces comparable to healthy subjects and with a forward dynamics comparable to the experimental training data, achieving an average correlation of 0.82 during stair ascent and of 0.58 during ramp ascent across both the knee and ankle joints.

## 1. Introduction

Deep reinforcement learning (DRL) is a powerful method to solve high-dimensional state-space problems. Thanks to this feature, DRL has been successfully used for the dynamic simulation of physics-based models of both humans [1,2,3,4] and robots [5,6,7] while performing different locomotion tasks. However, learning complex and physiologically plausible motions remains a research challenge in the simulation of physics-based musculoskeletal models [8].

With the long-term goal of using DRL for the design of the control architecture of lower-limb prostheses [9], this study proposed to use DRL to teach the physics-based musculoskeletal model of a human to ascend stairs and ramps, while achieving a forward dynamics and muscle fiber forces of the major muscles comparable to the ones of healthy subjects. The proposed method builds upon our previous work [4], in which we showed that DRL and, specifically, the Proximal Policy Optimization (PPO) algorithm in combination with imitation learning can teach physics-based musculoskeletal models of both able-bodied and impaired subjects to perform level-ground walking.

The proposed DRL method is sketched in Figure 1 and was developed in the open-source simulation software OpenSim [10] (NIH National Center for Simulation in Rehabilitation Research, Stanford, CA, USA, https://opensim.stanford.edu/, (accessed on 31 August 2021)). The agent is a physics-based human musculoskeletal model and is trained by a deep neural network to learn how to ascend stairs and ramps, which are represented as two objects in the simulation environment. The deep neural network receives, as inputs, the observed state (joint kinematics and muscle forces) of the agent and a reward, which is computed on an objective function and an imitation learning term. The output of the neural network is an action, namely, the muscles’ exitations of the agent.

The human model used in this study was modified from [1] and consists of 18 muscles (each leg has 6 uniarticular muscles and 3 biarticular muscles) to control 14 degrees of freedom. The model is kept as simple as possible to allow the DRL architecture to teach the model how to perform the tasks in a computationally efficient way but, at the same time, accurate enough to describe a healthy human subject. The model was scaled to fit the experimental data (belonging to the Motion Capture Database, which is provided by the Graphics Lab of the Carnegie Mellon University and publicly available in [11]), which are used as imitation data during training and for the validation of the proposed method. In order to explore locomotion tasks on advanced environments, such as ascending stairs and ramps, this study needed to substitute the Hunt–Crossely contact model (which can only be used with level ground) with the elastic foundation contact model at the feet of the human model and to add two meshes for the two different objects (the stairs or the ramp) in OpenSim.

To summarize, the main contributions of this paper are:To show that DRL, based on PPO in combination with imitation learning, can successfully teach a physics-based human musculoskeletal model in OpenSim to ascend stairs and ramps, with the future goal of using such architecture for the control of lower-limb prostheses.To be able to study more advanced environments in OpenSim, in addition to level ground, by implementing the elastic foundation model for the contact forces, as well as by introducing different objects’ meshes.

The remainder of the paper is organized as follows. Section 2 presents the physics-based human musculoskeletal model, the environment, and the used experimental data. Section 3 describes the proposed DRL method. Section 4 shows and discusses the results obtained during the simulations. Finally, concluding remarks are drawn in Section 5.

## 2. Materials

This Section presents the physics-based human musculoskeletal model and the objects (stairs and ramp) on which the model learns the respective gait patters and the experimental data from a public dataset that are used to train and validate the proposed DRL method.

### 2.1. Musculoskeletal Model

The 3D lower-extremity physics-based human musculoskeletal model used in this study was developed in OpenSim 3.3 (model version number 3000) as an .osim file. The model was provided for the NIPS’17 competition [1] as a simplification of the more complete and complex human musculoskeletal model in [12] for its easiness of use in combination with DRL algorithms. The model is composed of seven bodies: each leg has three bodies (an upper leg, a lower leg, and a foot) while the pelvis, torso, and head are represented by a single body (the angle between the pelvis and the torso is fixed to $-15^{\circ }$ for the stair ascending task and to $-5^{\circ }$ for the ramp ascending task). The model has 18 muscles (9 per each leg) to control 14 degrees of freedom. Each leg has 6 uniarticular muscles (i.e., gluteus maximus, iliopsoas, vastus lateralis, bicep femoris, soleus, and tibialis anterior) and 3 biarticular muscles (i.e., hamstring, rectus femoris, and gastrocnemius), as shown in Figure 2. The 14 degrees of freedom are distributed as follows: 6 at the pelvis (tilt, list, rotation, and translations in XYZ), 2 at each hip joint (flexion and adduction/abduction), 1 at each knee joint (flexion/extension), and 1 at each ankle joint (dorsiflexion/plantarflexion).

The muscles are modeled in OpenSim as Hill-type muscles, with a first-order dynamic between excitation and activation [13]. The generated muscle force is a function of the length, the velocity, and the activation level, ranging between 0% and 100%. When the muscles are activated, they generate a movement which is a function of the muscle properties (i.e., the maximum isometric force, the muscle fiber length, the tendon slack length, the maximum contraction velocity, and the pennation angle). It is important to notice that, to compensate for the reduced number of muscles in the used OpenSim model with respect to a complete human muscluloskeletal model, the maximum isometric force for all muscles was increased by 80% to allow the model to perform the task of ascending stairs, while the isometric force for all muscles was left at its default value to perform the task of ascending the ramp.

To use the model for the purpose of this study, some of its sections were modified. Table 1 summarizes the changes that were made, i.e., the objects (stairs and ramp) were designed and added to the BodySet section because only level ground is provided in OpenSim; spherical meshes were added to the feet in the ContactGeometrySet section to ensure proper contact forces between the feet and the stairs/ramp; the Hunt–Crossely model was replaced with the elastic foundation force model within the ForceSet section to allow for introducing contact forces on the new objects (the Hunt–Crossely model is not compatible with the new objects). These modifications are further detailed hereafter.

#### 2.1.1. Feet

The contact geometry of each foot was realized with three spherical meshes, i.e., one heel and two toes. The heel mesh is a sphere with a diameter of 50 mm, whereas each toe mesh is a sphere with a diameter of 25 mm. These three meshes are triangular-hollow (i.e., only with exterior faces) meshes, and their level of granularity is made of 107 vertices and 210 faces. Each mesh is designed to keep the amount of vertices and faces to a minimum to limit the computational load during simulations. Figure 3 shows the placement of the contact meshes with respect to the bones in each foot.

The heel/toes meshes are imported into the ContactGeometrySet section of the .osim model file. The coordinates (X, Y, Z) for the geometry of each foot are specified as follows: the heel is placed at (30, 20, 0) mm relative to the body piece calcn, and toes 1 and 2 are placed at (20, −5, −26) mm and (20, −5, 26) mm relative to the body piece toes, respectively.

During simulations, the ground reaction forces are exerted at these three contact meshes, but not at the bones. To ensure that the meshes are able to contact the object (stairs or ramp) and to generate the correct ground reaction forces, the contact dynamics cannot be described by the default Hunt–Crossley model, so it was substituted with the elastic foundation force model. The elastic foundation force model is required as a substitute as this force model is the only one that OpenSim accepts when dealing with custom geometry (stairs/ramp with heels and toes). The coefficients of the elastic foundation model are summarized in Table 2 and were selected as in [14]. Specifically, the stiffness coefficient was estimated on shoe rubbers and the static/dynamic/viscous friction coefficients were selected to avoid slippage during contact, whereas the dissipation coefficient to avoid bouncing at contact.

#### 2.1.2. Design of the Objects: Stairs and Ramp

Two objects were developed in this study, i.e., the stairs and the ramp. The geometry of the two objects are two different meshes, which were created by using OpenSim together with MeshLab 2020.07 (MeshLab, Pisa, Italy, www.meshlab.net (accessed on 31 August 2021)), SketchUp 2017 (Trimble Inc., Westminster, CO, USA, www.sketchup.com (accessed on 31 August 2021)), and Blender 2.83 (Blender, Amsterdam, The Netherlands, www.blender.org (accessed on 31 August 2021)), and saved as two different .obj files.

The two meshes are closed triangular-hollow (i.e., only with exterior faces) watertight meshes, and their level of granularity is made of 48 vertices and 72 faces for the stairs and of 24 vertices and 36 faces for the ramp. The meshes were designed to keep the amount of vertices and faces to a minimum to limit the computational load during simulations. Each mesh has no overlapping faces or duplicate vertices. The normal to each face point outwards, and they are directly calculated by OpenSim by considering the order (clockwise or anti-clockwise) in which the vertices are defined. The two meshes are imported into the BodySet section and the ContactGeometry section of the .osim model file as non-manifold watertight meshes.

The stairs are 2 m wide and are made of 3 steps. A single step has a height of 0.2 m and a depth of 0.25 m. The total increase in height is 0.6 m. The ramp is 2 m wide, has a run of 3.2 5m, and a rise of 0.45 m. The gradient of the slope is 7.883${}^{\circ }$, with a total length of 3.28 m. Note that the dimensions of the objects were chosen to match the stairs/ramp of the Carnegie Mellon University Graphics Lab motion capture public dataset [11], from which the training data were also taken.

Figure 4 show the two environments in which the model is at its starting position (corresponding to the starting position in the experimental dataset) in front of the stairs (Figure 4a) and the ramp (Figure 4b). The model is facing and performing the gait in the positive X direction. The left and right of the model are the negative and positive Z directions, respectively. The up and down motion of the model are the positive and negative Y directions, respectively. The reference for the center of the model is its pelvis, which is placed at (0,0.94,0) m in both environments.

### 2.2. Simulation Settings

A simulation runs in real time but produces data at each time-step throughout one iteration. In this paper a time-step is defined as the shortest amount of time between two events within the simulation. For example, with a time-step of 0.2 s it would be possible to activate a muscle every 0.2 s. An iteration is defined as a run of the simulation between t=0 s and whenever the simulation restarts. If the simulation has restarted three times, it is then starting its fourth iteration of the simulation. We recorded the data at each time-step for the very last iteration of the simulation. The simulation was manually halted under defined criteria. Once the reward gain was leveling off, the simulation was stopped. After the task was deemed successful, e.g., the model made it to the top of the stairs, it was also stopped. Lastly, it is possible that the simulation could keep learning, but we decided to end all simulations after a maximum of seven days.

### 2.3. Training Dataset

The data used in this study belong to the Carnegie Mellon University Graphics Lab Motion Capture Database and are publicly available in [11]. The motion capture data were collected on healthy subjects by using Vicon infrared MX-40 cameras (Vicon Motion Systems Ltd., Oxford, UK, www.vicon.com (accessed on 31 August 2021)) on 41 markers placed on the subjects’ bodies and saved in .c3d files. The ease of use, free availability, and wide range of possible tasks, as well as documentation, were our justification for using their data.

From the Carnegie Mellon University dataset, this study uses the motion capture data from the lower body of subject #14 (during the trial #22) for the stairs ascending task and of subject #74 (during the trial #19) for the ramp ascending task. The motion capture data of the lower body were exported to a .trc tracking file by using the Mokka 0.6.2 software (Motion Kinematic & Kinetic Analyzer, Lausanne, Switzerland, biomechanical-toolkit.github.io/mokka/ (accessed on 31 August 2021)).

The tracking data were used on the human model in OpenSim to run the inverse kinematics, from which the training data are derived for this study. The original Carnegie Mellon University data and the kinematic data generated through OpenSim report a maximum root mean squared error of 0.05 m for all markers, which is in agreement with the guidelines for the verification and validation of human musculoskeletal models in OpenSim [15].

For each joint *i* in the model, the velocity $v_{i}\left(t\right)$ at time *t* was calculated as follows:(1)$$v_{i}\left(t\right)=\frac{p_{i}\left(t\right)-p_{i}\left(t-1\right)}{t}$$
where $p_{i}\left(t\right)$ and $p_{i}\left(t-1\right)$ are the positions of the joint at time *t* and t−1, respectively. The initial velocity $v_{i}\left(0\right)$ is assumed to be equal to $v_{i}\left(1\right)$, while the final velocity $v_{i}\left(m\right)$ is assumed to be equal to $v_{i}\left(m-1\right)$, with *m* being the total amount of data in the dataset.

The simulation uses the task’s full dataset to train on and then to validated on after the simulation had ended. Even though the human gait is very cyclical, we chose to use an entire dataset to train and validate on rather than using repeating patterns due to (a) having the opportunity to use the entire gait data and (b) the tasks including starting and ending from stand still, which is included in the dataset (the model takes the first step with the left leg for both the stair and ramp gait initiations).

## 3. Method

This Section describes the DRL algorithm that is used to teach the human musculoskeletal model to ascend the stairs and the ramp, as sketched in Figure 1.

### 3.1. Deep Neural Network

The neural network designed in this study is a multi-layer perceptron, i.e., a feed-forward artificial neural network, that consists of four layers. Specifically, the input layer has 214 neurons, both hidden layers have 312 neurons, and the output layer has 18 neurons. For each neuron $\nu_{i}$, the output *y* is calculated by using a general output function, i.e.,
(2)$$y\left(\nu_{i}\right)=\tanh\left(b+\sum_{i=1}^{n}x_{i}w_{i}\right)$$
where *x* is the input to the neuron, *w* is the weight between the current and previous neuron, *b* is the bias, *n* is the number of inputs from the previous layer, and tanh is the activation function.

The input of the neural network is the state $s_{t}$, which is a 214-dimensional vector of continuous variables. Specifically, the kinematic variables are: the positions and rotations, the linear and rotational velocities, and the linear and rotational accelerations of the joints and body segments. The force variables are: the ground reaction forces, the muscle forces, the muscle fiber lengths and velocities, the tendon forces, and the additional miscellaneous forces that impose limits to the muscle forces. Table 3 summarizes the complete set of state variables for the musculoskeletal model.

The output of the neural network is an action $\alpha_{t}$, i.e., an 18-dimensional vector, where each variable represents either a 1 or a 0 that indicates the muscle excitations. The output vector is fed into the OpenSim predefined *brain-controller* that, from the muscle excitations, calculates the muscle activations by using the first-order dynamics equations of the Hill-type muscle model.

### 3.2. The Learning Algorithm: PPO

The learning algorithm used in this study is PPO [16], and the hyper-parameters for the network were taken from our previous work [4]. During training of the neural network, the weights $w_{i}$ of the connections between the neurons are optimized by PPO, such that the network outputs desirable actions $\alpha_{t}$ at time *t* based on the observed state $s_{t}$ at time *t*. Specifically, PPO uses the following objective function:(3)$$L^{CLIP}\left(\theta \right)={\hat{E}}_{t}\left[min\left(r_{t}\left(\theta \right){\hat{A}}_{t},clip\left(r_{t}\left(\theta \right),1-\epsilon ,1+\epsilon \right){\hat{A}}_{t}\right)\right]$$
where ${\hat{E}}_{t}$ is the expected value at time *t*, ${\hat{A}}_{t}$ is the advantage estimation at time *t* (i.e., the difference between the expected and the real reward from an action), ϵ is the clip value, and $r_{t}\left(\theta \right)$ is the ratio at time *t* between the probabilities of the new policy $\pi_{\theta }$ in the parameters θ and the old policy $\pi_{\theta_{old}}$ in the parameters $\theta_{old}$, i.e.,
(4)$$r_{t}\left(\theta \right)=\frac{\pi_{\theta }\left(\alpha_{t}|s_{t}\right)}{\pi_{\theta_{old}}\left(\alpha_{t}|s_{t}\right)}$$
and the clipping term $clip\left(r_{t}\left(\theta \right),1-\epsilon ,1+\epsilon \right){\hat{A}}_{t}$ negates the incentive for the rate $r_{t}$ to move outside of the clip-bound [1−ϵ,1+ϵ] set by ϵ.

### 3.3. Reward Function

The action $\alpha_{t}$ generated by the model results in a reward, which can be either positive or negative depending on the action the model just took. If the model does a favorable action, it receives a positive reward. Otherwise, if the model does an unfavorable action, it receives a negative reward, i.e., a penalty. To learn the tasks, the model should receive as many rewards as possible.

In this study, as also proposed in our previous work [4], the reward function $J{\left(\pi \right)}_{t}$, calculated at each time-step *t* consists of two parts, i.e., a goal reward $J_{goal}{\left(\pi \right)}_{t}$ and an imitation reward $J_{imitation}{\left(\pi \right)}_{t}$, which are weighted as:(5)$$J{\left(\pi \right)}_{t}=0.1\cdot J_{goal}{\left(\pi \right)}_{t}+0.9\cdot J_{imitation}{\left(\pi \right)}_{t}$$

The goal reward is:(6)$$J_{goal}{\left(\pi \right)}_{t}=e^{-8\cdot \sum_{t}\left(p_{vel}\right)}$$
where $p_{vel}$ is the penalty for deviating from the desired pelvis velocity contained in the imitation data.

The imitation reward is:(7)$$J_{imitation}{\left(\pi \right)}_{t}=0.9\cdot e^{-4\cdot \sum_{t}p_{pos}}+0.1\cdot e^{-0.1\cdot \sum_{t}p_{vel}}$$
At each time-step *t*, both the position penalty $p_{pos}$ and the velocity penalty $p_{vel}$ of the pelvis, hip, knee, and ankle joints are calculated. The penalties are calculated by taking the sum of the squared error of the difference between what is observed and what is in the dataset at time *t*.

### 3.4. Implementation

To run the simulations, different mediums of hardware were used. The most prominent is the Microsoft Azure cloud computing services (azure.microsoft.com (accessed on 31 August 2021)). A total of three servers were allocated on Azure, i.e., three NC6 Data Science Virtual Machine for Linux (Ubuntu 18.04) with 6 cores and 56 GB of RAM. Other hardware was used as well, i.e., four systems running Ubuntu 18.04, i.e., system 1 has an Intel core i5 3570K and 8 GB of RAM, system 2 has an Intel core i7 8700K with 16 GB of RAM, system 3 has an AMD Ryzen 3800x with 32 GB of RAM, and system 4 has an AMD Ryzen 5950x with 64 GB of RAM.

The version of Python used is 3.6.10 (www.python.org (accessed on 31 August 2021)), and the version of Tensorflow is 1.15 (www.tensorflow.org (accessed on 31 August 2021)). The DRL algorithm makes use of mpi4py to be able to run on multiple cores (usually between 4–16).

The approximate time to run a single iteration on the Azure servers is 356 s, on system 1 is 250 s, on system 2 is 170 s, system 3 is 112 s, and system 4 is 95 s.

Data are collected from the simulation after each iteration. The rate at which the data are sampled during the experiment is every 5 iterations. The trained model is saved using tensorflow-checkpoint. The angles (in degrees) of the joints are extracted from the trained model. The training reward per time-step is also recorded as the mean of the reward. The length of each training period is recorded as well.

## 4. Results and Discussion

This Section presents and discusses the results obtained in the simulations (according to their run-time) of the human model while learning to ascend the stairs and the ramp by means of the proposed DRL architecture. The results are also shown in the video that is provided as Appendix A to this study.

### 4.1. Stairs Ascent

Figure 5 shows the reward obtained by the model while learning to ascend the stairs. The reward increases rapidly for the first ∼19,383 time-steps, then it levels out until ∼55,951, where it then again increases rapidly until ∼76,099; from here onward, the learning is continuous and slow. The rapid increases and jumps in the reward represent when the model has learned to correctly take the steps. In the end, the model is able to climb three steps of the stairs for ∼300 time-steps and reaches a maximum reward of ∼215. The model is able to achieve 71.6% of its total reward, suggesting that the gait is 71.6% accurate to the imitation data.

Figure 6 shows the left/right knee (top) and ankle (bottom) joint angles for the entire simulation time, which includes ascending three steps. It can be noted that there is a good correlation between the experimental data (dotted lines) and the forward dynamic simulation (continuous lines). Specifically, the left and right knee joints show a correlation of above 0.9, while the left and right ankle joint of above 0.6. From the figure, it can be observed that the left leg initiates the gait of ascending the stairs by flexing the knee. At the same time, to balance the forward progression, the right leg is maintained in a steady extension until the weight acceptance of the left leg. From 0.9 s to 1.2 s, the pull-up phase of the left leg occurs, where the body is lifted-off from the step. This behavior can be confirmed by observing that the right knee is clearly off the ground, and it flexes to provide the ground clearance to step over to next step. This cyclic pattern is very stable in the simulation and, after successfully ascending the three steps, the model attempts to take the next step but, due to the unavailability of the additional steps in the object, the model falls down and the simulation is terminated. This can be noted by the discrepancies between the imitation and experimental data towards the end of simulation. Similarly for the ankle, it is interesting to note the stability of the left ankle joint to maintain the weight acceptance, pull-up, and forward continuance phase between 0.9 s and 1.7 s. Moreover, from 1.7 s to 2.4 s, the ankle quickly plantarflexes to provide toe-clearance along with knee flexion for the correct foot placement over the next step.

Figure 7 shows the major muscle contributors of both legs for the stair ascent task, i.e., bicep femoris short head (bifmesh), vasti, soleus, and tibialis anterior. The bifmesh is a knee flexor muscle, and it also provides support to the overall gait together with the hamstrings. The average force of the bifmesh is 6100 N and 6500 N for the left and right leg, respectively. From the figure, it can be observed that the bifmesh is triggered throughout the gait to maintain the stability of the knee joint. Moreover, during every heel strike, there is a peak force of about 1200 N to resist the impact and achieve the weight acceptance. The vasti is a knee extensor muscle and maintains an average force of 2700 N and 2400 N for the left and right leg, respectively. From the figure, the cyclic pattern of the vasti during the weight acceptance phase is clear for both legs. Moreover, once the knee is extended in the pull-up phase, the vasti slowly relaxes during the forward progression as the center of body mass is pushed forward to take the next step. The soleus and the tibialis anterior are responsible for the ankle plantarflexion and dorsiflexion, respectively. The soleus is activated to provide the peak muscle forces during the end of the forward-continuance phase (from 0.9 s to 2.5 s) for the right foot and to provide enough push-off force to the leg so to lift the body to take the next step. The tibialis anterior is triggered especially to accept the weight and to provide toe clearance after heel strike. Finally, from the figure, it can be noted that the bifmesh and the tibialis anterior exceed the nominal maximum isometric forces when compared with the vasti and soleus. This could be due to the reduced muscle model that is used in this study. For instance, in the case of simulations with a model with more muscles, the arrangement of muscles such as gastrocnemius and semimembranosus could reduce the forces of the bifemsh, while the extensor digitorum muscles could support the tibialis anterior during ankle dorsiflexion.

Figure 8 shows the ground reaction forces in the Y direction, computed at the zero moment point of each foot during two complete gaits (i.e., each foot is in contact with the ground twice and is lifted twice). The Y-component shows a peak at the very beginning and at the end of the contact. It maxes out at a force of −1602 N on the left and −1387 N on the right leg. It has an average (not including zeros) of −500 N on the left leg and −580 N of the right leg.

### 4.2. Ramp Ascent

Figure 9 shows the reward obtained by the model while learning to ascend the ramp. The reward increases rapidly for the first ∼13,000 time-steps, then it increases linearly. The initial increase in the reward is due to the model learning to stand and to take the first step. Since the environment is continuously increasing after this initial step, the reward exhibits continuous learning. The further the model walks up the ramp, the higher the reward it achieves. The final 4000 time-steps show the greatest fluctuation. This part of the task is only learned at a later stage as it takes time to reach the top of the ramp, resulting in the final moments being iterated over less time. The model spends most of its training time at the beginning of the ramp as this is where it resets to. In the end, the model was able to reach the top of the ramp after ∼210 time-steps, reaching a maximum reward of ∼120. The model was able to achieve 57.1% of its total reward, suggesting that the model’s gait is 57.1% accurate to the imitation data.

Figure 10 shows the left/right knee (top) and ankle (bottom) joint angles for the entire simulation time. It can be noted that there is some correlation between the experimental data (dotted lines) and the forward dynamic simulation (continuous lines). Specifically, the left and right knee joints show a correlation of above 0.6, while the left and right ankle joints show a lower correlation. From the figure, it can be noted that both the knee joints do not perform a complete extension, which is in agreement with the biomechanics of ramp ascent for able-bodied subjects [17]. The right knee joint performs a faster flexion during the swing phase (from 1 s to 1.5 s) compared with the imitation data to provide foot clearance, while the left foot continues from mid-stance to toe-off. Moreover, both knees show a cyclic pattern that is in agreement with the ramp ascending tasks. The ankle is in dorsiflexion for both legs. However, the left ankle remains at 22${}^{\circ }$ throughout the simulation. The left ankle tries to provide a push-off at 1.7 s, right before the swing phase, but it is not as significant as observed in the right ankle. From the figure, it seems that the right ankle works more than the left one. Both ankles show a cyclic pattern, and their kinematics are consistent with the experimental data of healthy subjects. The maximum dorsiflexion and plantarflexion in the ankle is observed to be around 25${}^{\circ }$ and −10${}^{\circ }$, respectively. However, the proposed DRL method enables the human model to learn the task.

Figure 11 shows the major force contributors for the ramp ascent task, i.e., bicep femoris short head (bifmesh), vasti, soleus, and tibialis anterior. The bifmesh generates an average of 880 N and 930 N for the left and right leg, respectively. The bifmesh is activated throughout the gait with peak in contribution during the mid-to-late stance phase of the gait to support in the active flexion of the knee joint and to provide support to the hamstring muscles. The vasti shows a very clear force contribution on the left leg compared with the right leg, with an average force of 1310 N and 1330 N, respectively. The vasti is activated at the heel strike of the foot to help in the knee extension. However, in the right foot, the vasti is also triggered during the swing phase, mainly to provide stability and for the positioning of the foot for the next step. Similarly, the soleus also shows a distinctive force contribution that matches with the biomechanics of the healthy human gait. The soleus is activated only during the push-off phase of the gait for both the ankle joints to achieve plantarflexion. The average forces applied to the joints are 1810 N and 1740 N for the left and right foot, respectively, which are below the maximum isometric forces. The tibialis anterior contributes with an average force of 2120 N and 2000 N for the left and right foot, respectively, to achieve the dorsiflexion throughout the gait. The agonistic behavior of the tibialis anterior is not seen during the gait as it is also triggered during the push-off phase, which would make the soleus muscle work even harder to overcome this force. However, to perform a successful ramp ascent, the toe-clearance is very critical and, to enable this functionality, the DRL method learns to keep the tibialis anterior activated.

Figure 12 shows the ground reaction forces in the Y direction, computed at the zero moment point of each foot during ramp ascent (i.e., each foot is in contact with the ground four times and is lifted three times). The Y-component often shows peaks. It maxes out at a force of −2668 N on the left leg and −2469 N on the right leg. It has an average (not including zero values) of −527 N on the left leg and of −616 N of the right leg.

### 4.3. Evaluation

The human musculoskeletal model could learn to ascend both stairs and the ramp. Table 4 reports the correlation values of the knee and ankle joints for the two tasks computed on the simulation but excluding the last 0.5 s (i.e., the part of the simulation when the model falls because the object ends). The model scores a lower correlation value during the ramp ascent task (0.58) compared with the stair ascent task (0.82). It should be noted that the right ankle has the lowest correlation in the stairs ascent task. The reason is that the right foot is only in contact with the ground once during this task, leaving it not in contact for most of the simulation. The angle of the foot, while not in contact, is less important for learning the task, resulting in a low correlation.

## 5. Conclusions

This paper proposed to use DRL to teach a physics-based human musculoskeletal model to ascend stairs and ramps. The method is based on the PPO in combination with imitation learning and is implemented in OpenSim.

Compared with our previous work [4], in which a similar method was used for level-ground walking, this study showed that PPO with imitation learning is able to cope with more complex environments, such as stairs and ramps, on which the model can learn forward dynamics comparable to the experimental training data, achieving a correlation of 0.82 during stair ascent and of 0.58 during ramp ascent across both the knee and ankle joints. The muscle forces are comparable among the two legs (almost no asymmetry is observed), and their average values are compatible with values from healthy human subjects, suggesting the biomechanical accuracy of the musculoskeletal simulation. However, with added computational power and time, adding upper body joints and muscles, as well as arms, would be of interest.

The DRL method, together with the introduction of a novel reward function, the elastic foundation model for the contact forces, and the meshes for the generation of novel environment in OpenSim, built a solid base for future research into the analysis of human locomotion on complex terrains and, possibly, into the control of lower-limb prosthetic devices. 

## Figures and Tables

**Figure 1 sensors-22-08479-f001:**
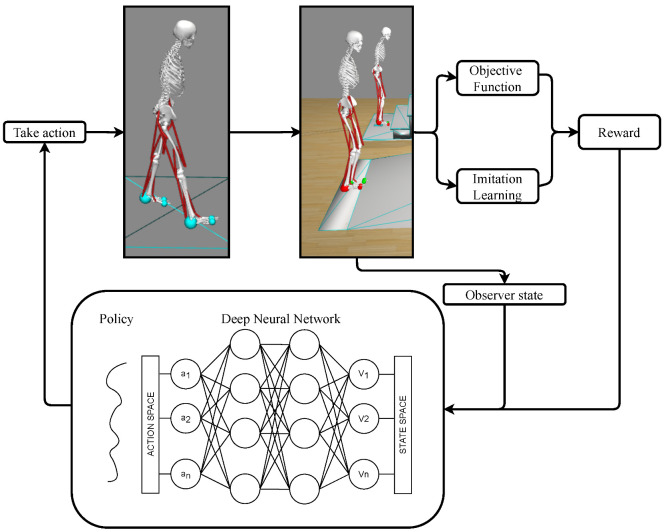
The proposed DRL method for the dynamic optimization of the forward dynamics of a human musculoskeletal model during stairs or ramp ascent.

**Figure 2 sensors-22-08479-f002:**
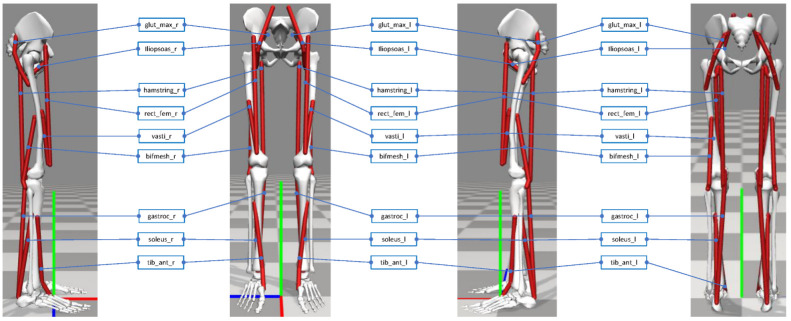
The physics-based human musculoskeletal model developed in this study. Figures from left to right: side view facing the right leg, front view, side view facing the left leg, and back view.

**Figure 3 sensors-22-08479-f003:**
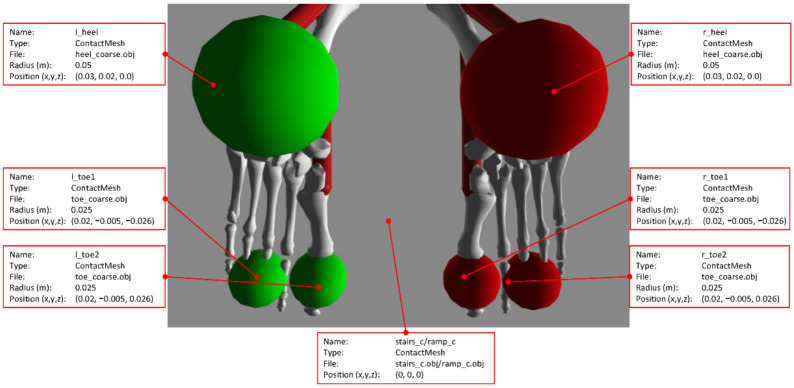
Bottom view of the spherical contact meshes for the heels and toes with respect to the bones in the feet (left foot in green, right foot in red, and feet bones in white).

**Figure 4 sensors-22-08479-f004:**
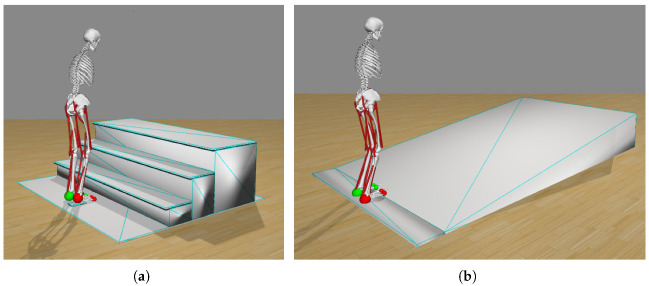
Simulation environments: Objects and human model at the starting position of the experimental dataset. (**a**) Stairs (with three steps). (**b**) Ramp.

**Figure 5 sensors-22-08479-f005:**
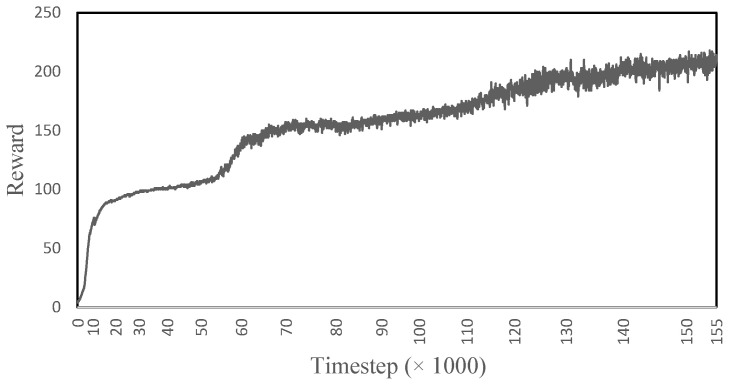
The reward obtained during the learning process of the human musculoskeletal model to ascend the stairs.

**Figure 6 sensors-22-08479-f006:**
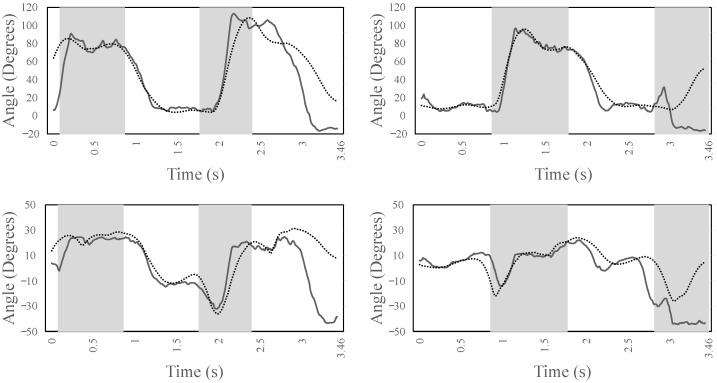
The kinematics of the left/right knee (**top**) and left/right ankle joint (**bottom**) during stair ascent for the entire simulation time. The areas marked in gray show the period where the leg is not in contact with the ground. The dotted lines are the experimental data; the continuous lines are the forward dynamic simulation data.

**Figure 7 sensors-22-08479-f007:**
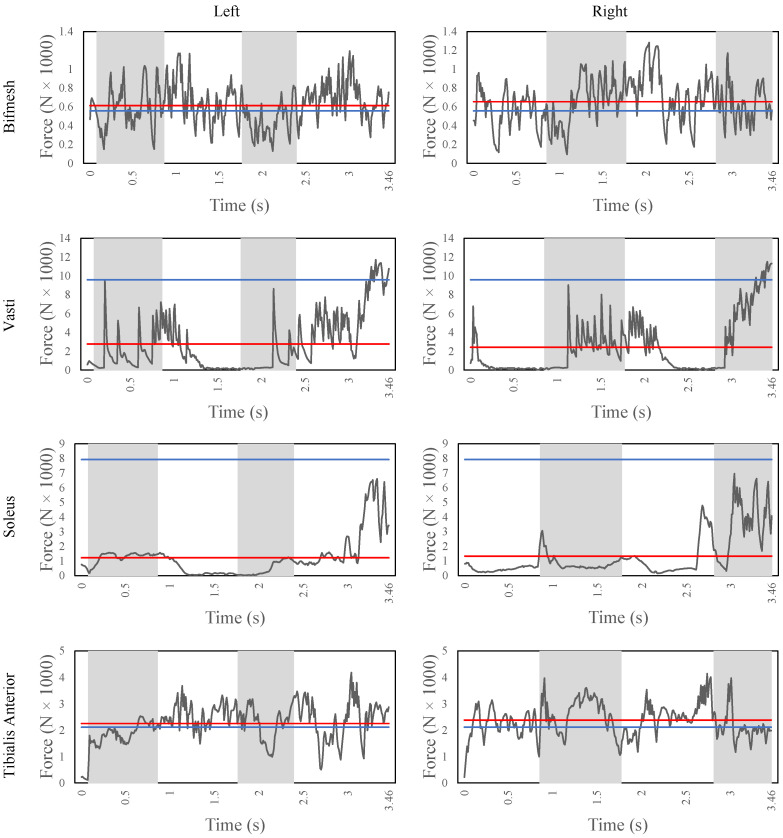
The muscle fiber forces of the biceps femoris, vasti, soleus, and tibialis anterior (from top to bottom) of the human model during stair ascent for the entire simulation time. The figures on the left side refer to the left leg, while the ones on the right side refer to the right leg. The horizontal red line indicates the mean fiber force, while the horizontal blue line indicates the maximum isometric forces.

**Figure 8 sensors-22-08479-f008:**
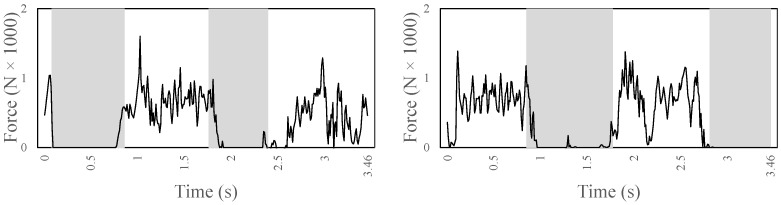
The ground reaction force in the Y direction computed at the zero moment point of each foot (left foot on the left and right foot on the right) during stair ascent.

**Figure 9 sensors-22-08479-f009:**
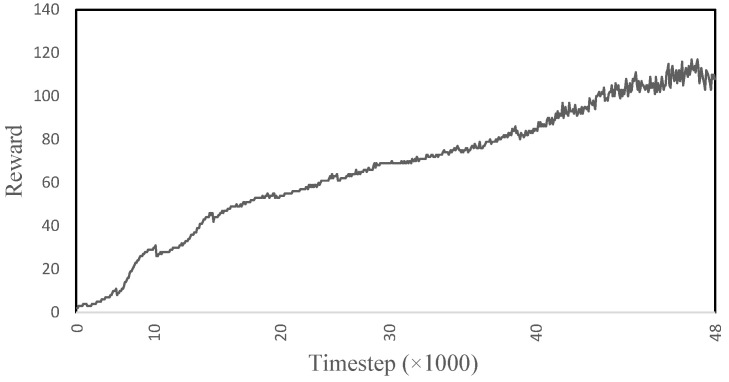
The reward obtained during the learning process of the human musculoskeletal model to ascend the ramp.

**Figure 10 sensors-22-08479-f010:**
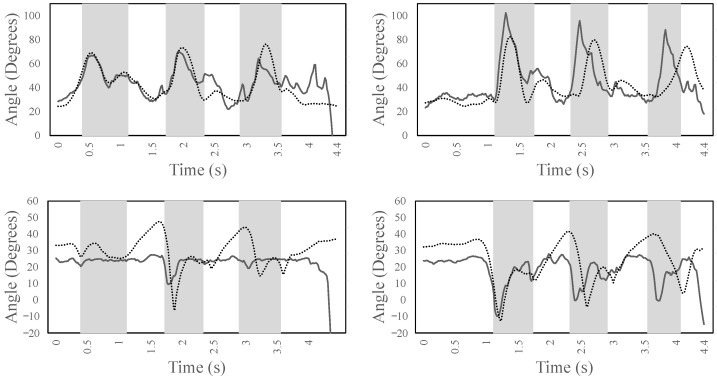
The angles of the left/right knee (**top**) and of the left/right ankle joint (**bottom**) during ramp ascent for the entire simulation time. The areas marked in gray show the period where the leg is not in contact with the ground. The dotted lines are the experimental data; the continuous lines are the forward dynamic simulation data.

**Figure 11 sensors-22-08479-f011:**
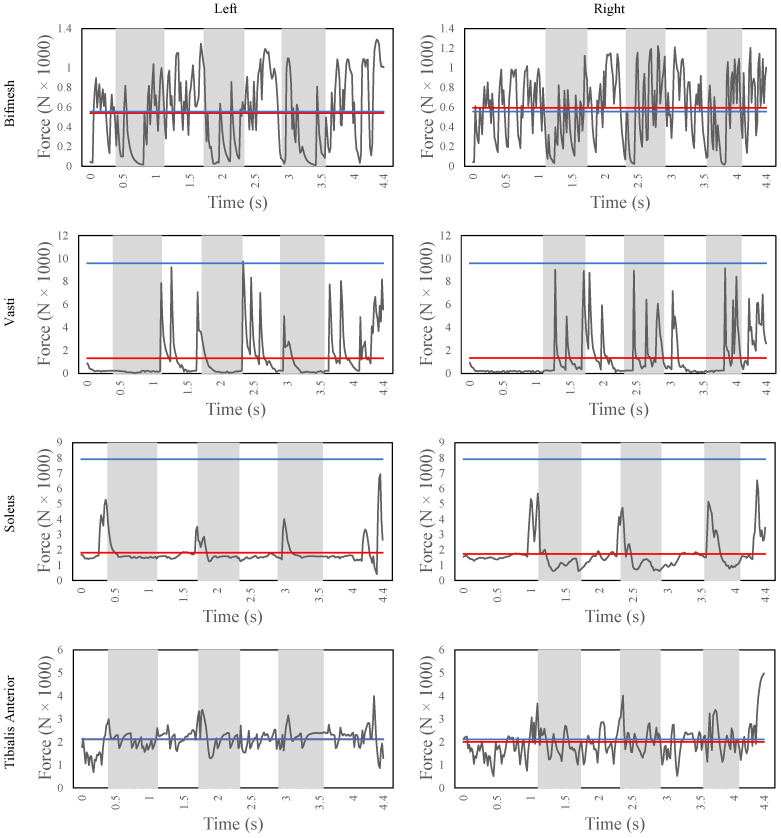
The muscle fiber forces of the bicep femoris, vasti, soleus, and tibialis anterior (from top to bottom) of the human model during ramp ascent for the entire simulation time. The figures on the left side refer to the left leg, while the ones on the right side refer to the right leg. The horizontal red line indicates the mean fiber force, while the horizontal blue line indicates the maximum isometric forces.

**Figure 12 sensors-22-08479-f012:**
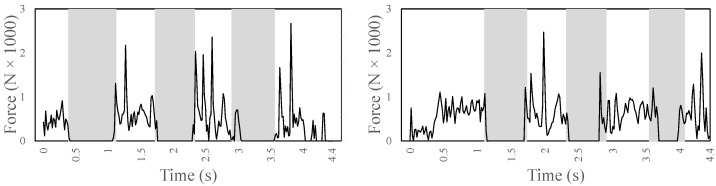
The ground reaction force in the Y direction computed at the zero moment point of each foot (left foot on the left and right foot on the right) during ramp ascent.

**Table 1 sensors-22-08479-t001:** The sections in the .osim file of the model developed in this study: some sections are left to their default state, others are modified.

Section	Description	Modification
BodySet	Body geometry	Addition of objects
ConstraintSet	List of constraints	-
ForceSet	Acting forces	Elastic foundation
MarkerSet	List or markers	-
ContactGeometrySet	Contact geometry	Spherical feet meshes
ControllerSet	Auxiliary controllers	-
ComponentSet	Group geometry	-
ProbeSet	Auxiliary probes	-

**Table 2 sensors-22-08479-t002:** The coefficients of the elastic foundation model between the feet and the objects. The geometry value for the object can be chosen from either one of the two objects (i.e., stairs_c or ramp_c), while for the right (_r) or left (_l) foot the geometry values are identical.

	Coefficient	Value
Right leg	appliesForce	true
	geometry	[stair_c, ramp_c] r_heel r_toe1 r_toe2
	dissipation [s/m]	5
	stiffness [MPa/m]	50
	static_friction	0.9
	dynamic_friction	0.9
	viscous_friction	0.9
	transition_velocity	0.1
Left leg	appliesForce	true
	geometry	[stair_c, ramp_c] l_heel l_toe1 l_toe2
	dissipation [s/m]	5
	stiffness [MPa/m]	50
	static_friction	0.9
	dynamic_friction	0.9
	viscous_friction	0.9
	transition_velocicty	0.1

**Table 3 sensors-22-08479-t003:** The state variables of the human musculoskeletal model.

	n. of Variables
Positions/Rotations of body segments	13 + 13
Linear/Rotational velocities of the body segments	13 + 13
Linear/Rotational accelerations of the body segments	13 + 13
Positions/Velocities/Accelerations of the joints	17 + 17 + 17
Muscle forces	72
Miscellaneous forces	13
Total size of the state vector	214

**Table 4 sensors-22-08479-t004:** The correlation value for each joint during stair and ramp ascent.

	Stairs Ascent	Ramp Ascent
Left knee	0.92	0.84
Right knee	0.98	0.61
Left ankle	0.92	0.36
Right ankle	0.57	0.52
Mean	0.82	0.58

## Data Availability

Data-set is available at http://mocap.cs.cmu.edu/ (accessed on 31 August 2021).

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
