# Peer review of "Learning to Ascend Stairs and Ramps: Deep Reinforcement Learning for a Physics-Based Human Musculoskeletal Model"

_sensors, 2022, doi:10.3390/s22218479_

Round 1
Reviewer 1 Report
This paper introduces to a DRL approahto teach a physics-based human musculoskeletal model to ascend stairs and ramps.
The paper is very well presented in terms of formal analysis discussion and conclusions.
However in the part of the introduction and validation and methology it is sad that a gait dataset has been used but no reference or details are given on that. It is necessary to add biomechanical movement analysis state of art.
Adding and referencing these papers paper detailing the gait parameters will strengthen the soundness of the paper and such details can be used to improve even more the discussion:
https://doi.org/10.1007/978-3-319-09411-3_62
After this very minor revision the paper is ready in my opinion for publication due to the value of the work.
Reviewer 2 Report
This article presents use of a deep reinforcement learning algorithm to teach a physics-based human computational model focussed on kinetics and kinematics of the lower-limbs. The model is analysed in two conditions; namely, ascending stairs, and ascending a ramp. Model is built in and governed by OpenSim software. The research study seems to be technically sound. The manuscript is well-structured and well-written. Authors may wish to address the following comments to further improve the quality of the paper:
1. The particulars of the elastic foundation model used need clarification. As detailed in Table 2, static and dynamic friction coefficients are the same and both 0.9, units for stiffness are required, also the reasons for the selected values for dissipation and viscous friction can be elaborated.
2. The OpenSim model appears to embody a limited upper body definition, devoid of arms. As this can be reasonable for simplification purposes, in a dynamic study where the forces are the output, the mass characteristics of the upper body (where it is lumped, if so), as well as the dynamics motion of the arms have significant potential to alter the force output. An explanation regarding how the upper body is handled and its contributions are reflected would add value to the paper.
3. In a mostly symmetrical (wrt sagittal plane) model as such, why is the correlation value for the right ankle (0.57) during stairs ascent significantly lower than the other values (averaging 0.94) for the same case (as shown in Table 4). An elaboration on this matter - maybe in Section 4.3 - would be useful.
